# *PKD1L1* Is Involved in Congenital Chylothorax

**DOI:** 10.3390/cells13020149

**Published:** 2024-01-12

**Authors:** Jonathan B. Whitchurch, Sophia Schneider, Alina C. Hilger, Ricarda Köllges, Jil D. Stegmann, Lea Waffenschmidt, Laura Dyer, Holger Thiele, Bhanupriya Dhabhai, Tikam Chand Dakal, Andreas Müller, Dominic P. Norris, Heiko M. Reutter

**Affiliations:** 1Mammalian Genetics Unit, MRC Harwell Institute, Harwell Campus, Oxfordshire OX11 0RD, UK; jonathanwhitchurch@gmail.com (J.B.W.); l.dyer@har.mrc.ac.uk (L.D.); dominic_paul_norris@tiscali.co.uk (D.P.N.); 2Department of Neonatology and Paediatric Intensive Care, University Hospital Bonn Center of Paediatrics, 53127 Bonn, Germany; sophia.schneider@uni-bonn.de (S.S.); ricarda.koellges@posteo.de (R.K.); jil.stegmann@uni-bonn.de (J.D.S.); a.mueller@ukbonn.de (A.M.); 3Institute of Human Genetics, University Hospital Bonn, 53127 Bonn, Germany; lea.waffenschmidt@uk-erlangen.de; 4Department of Pediatrics and Adolescent Medicine, University Hospital Erlangen, 91054 Erlangen, Germany; alina.hilger@uk-erlangen.de; 5Division of Neonatology and Pediatric Intensive Care, Department of Pediatrics and Adolescent Medicine, University Hospital Erlangen, 91054 Erlangen, Germany; 6Cologne Center for Genomics, University of Cologne, 50931 Cologne, Germany; holger.thiele@uni-koeln.de; 7Genome & Computational Biology Lab, Department of Biotechnology, Mohanlal Sukhadia University, Udaipur 313001, India; bhanupriyadhabhai@gmail.com (B.D.); tc.dakal@mlsu.ac.in (T.C.D.)

**Keywords:** *PKD1L1*, chylothorax, congenital, pleural effusion

## Abstract

Besides visceral heterotaxia, *Pkd1l1* null mouse embryos exhibit general edema and perinatal lethality. In humans, congenital chylothorax (CCT) is a frequent cause of fetal hydrops. In 2021, Correa and colleagues reported ultrarare compound heterozygous variants in *PKD1L1* exhibiting in two consecutive fetuses with severe hydrops, implicating a direct role of *PKD1L1* in fetal hydrops formation. Here, we performed an exome survey and identified ultrarare compound heterozygous variants in *PKD1L1* in two of the five case–parent trios with CCT. In one family, the affected carried the ultrarare missense variants c.1543G>A(p.Gly515Arg) and c.3845T>A(p.Val1282Glu). In the other family, the affected carried the ultrarare loss-of-function variant (LoF) c.863delA(p.Asn288Thrfs*3) and the ultrarare missense variant c.6549G>T(p.Gln2183His). Investigation of the variants’ impact on PKD1L1 protein localization suggests the missense variants cause protein dysfunction and the LoF variant causes protein mislocalization. Further analysis of *Pkd1l1* mutant mouse embryos revealed about 20% of *Pkd1l1^−/−^* embryos display general edema and pleural effusion at 14.5 dpc. Immunofluorescence staining at 14.5 dpc in *Pkd1l1^−/−^* embryos displayed both normal and massively altered lymphatic vessel morphologies. Together, our studies suggest the implication of *PKD1L1* in congenital lymphatic anomalies, including CCTs.

## 1. Introduction

Congenital chylothoraces (CCTs) belong to the congenital lymphatic anomaly spectrum. Intrauterine, chyle accumulates in the pleural cavity, the fetal lung is compressed, and venous blood flow to the heart is restricted, resulting in fetal hydrops. Frequently, affected fetuses die intrauterine or shortly after birth. In surviving infants, loss of chyle-soluble fluid leads to malnutrition, thrombophilia, and immunodeficiency [1]. CCTs affect about one in 10,000 fetuses with a described male-to-female ratio of 2:1 [2]. While the overall birth prevalence is rare, CCT is the most common cause of congenital pleural effusion [3]. CCT treatment is primarily symptomatic [1]. Infants may require mechanical ventilation or support by continuous positive airway pressure, inotropic medications, vasopressors, and pleural drainage. Dietary measures comprise formula milk that is low in long-chain and high in medium-chain triglycerides [3]. Several pharmacological agents have been used in the direct symptomatic treatment of CCT, aiming to reduce thoracic lymphatic flow. These agents comprise the somatostatin analog octreotide [4]; etilefrine, a sympathomimetic drug that causes contraction of smooth muscle fibers [4]; propranolol, a nonselective beta-receptor antagonist [5]; sildenafil, a phosphodiesterase-5 (PDE5) inhibitor [6]; and sirolimus, an inhibitor of the mammalian target of rapamycin (mTOR) [7]. In a substantial number of cases, pre- and perinatal treatment is not successful, and the course remains difficult with long-term hospital treatment.

To date, the origin of primary CCT remains elusive, and the few known genetic causes are heterogeneous, including chromosomal disorders, e.g., Down’s syndrome or Turner syndrome [8], or rare lymphatic disorders comprising CCTs within their phenotypic spectrum. In this context, about 20 proteins in the VEGFR-3 signaling cascade have been associated with the expression of congenital lymphatic disorders comprising CCTs [9]. The VEGFR-3 downstream signaling cascade involves the RAS/MAPK and PI3K/AKT pathways. Here, several of the known Noonan syndrome-associated RASopathies present with lymphatic disorders, including CCTs [10].

Hitherto, the polycystic kidney disease (PKD) protein family, also known as the Transient Receptor Potential Polycystin (TRPP), polycystin, or PC proteins [11], has not been within the focus of CCT research.

In 2021, Correa and colleagues reported two fetuses in the same family with ultrarare compound heterozygous deleterious variants in *PKD1L1* exhibiting fetal hydrops. With no other cause identified, Correa and colleagues assumed a direct involvement of the *PKD1L1* variants in fetal hydrops formation. Here, we surveyed the exome of five case–parent trios with CCT and identified ultrarare compound heterozygous variants in two affected individuals. In the first family, the affected carried two ultrarare missense variants c.1543G>A(p.Gly515Arg) and c.3845T>A(p.Val1282Glu). In the second family, the affected carried the ultrarare loss-of-function variant (LoF) c.863delA(p.Asn288Thrfs*3) and the ultrarare missense variant c.6549G>T(p.Gln2183His). *PKD1L1* encodes for polycystic kidney disease 1-like 1 (Uniprot: Q8TDX9), an 11-transmembrane protein forming a heteromeric calcium channel (PKD1L1/PKD2L1) in primary cilia [12]. *PKD1L1* has been associated with autosomal recessive visceral heterotaxia (MIM: 517205), a developmental disorder of left–right patterning presenting with visceral situs inversus and complex congenital heart malformations [13]. Besides visceral heterotaxia, *Pkd1l1* null mouse embryos present with edema and perinatal lethality in a substantial number of cases [14]. PKD1L1 largely consists of a long N-term ectodomain and a C-term 11-transmembrane polycystic kidney disease (PKD) channel domain. The N-term ectodomain contains a polycystin cation channel (PCC) domain, a “receptor for egg jelly” (REJ) domain, and a polycystin-1, lipoxygenase, alpha-toxin (PLAT) domain [15]. The mouse rks mutation site is also located there, suggesting that this N-term ectodomain is functionally significant. Within the PKD gene family, *PKD1* and *PKD2* were first identified as autosomal dominant PKD genes [16]. PKD1 and PKD2 act together in the primary cilia of the renal epithelium, forming a channel complex. PKD1, containing an extracellular domain, is thought to mechanically sense urinary flows, whereas PKD2 mediates Ca^2+^ influx as a channel [16,17]. While *Pkd2* knockout mice show organ laterality defects [18], *Pkd1* knockout mice exhibit normal left–right determination, and PKD1 is not found in mouse node cilia [19]. In general, polyhydramnios, severe edemas, and hemorrhages are universal findings in the homozygous mutants of either locus by 12.5 dpc [20,21]. In this context, Kim et al. demonstrated a primary role of *PKD1* mutations in vascular fragility [20]. Mouse embryos homozygous for the mutant allele (Pkd1L) exhibited subcutaneous edema, vascular leaks, and the rupture of blood vessels, culminating in embryonic lethality at 15.5 dpc. Support for a primary role of PKD proteins in vascular integrity was demonstrated by Garcia-Gonzalez and colleagues [22]. They crossed a floxed allele of *Pkd1* (Pkd1cond/cond) with *Pkd1^+/−^*, *Meox2-Cre^+^* mice to produce embryos in which the floxed *Pkd1* allele was inactivated in 6.5 dpc embryos but not in the placental trophoblasts or extra-embryonic endoderm lineages [23,24]. *Pkd1^cond/−^*, *Meox2-Cre^+^* mice also exhibited cystic kidneys, edema, and polyhydramnios, but a substantial fraction survived to birth [22]. Those that survived died shortly after birth of apparent respiratory failure. The hearts of these edematous *Pkd1^cond/−^*, *Meox2-Cre^+^* littermates appeared normal with intact valves and septa, making this organ an unlikely cause of their edema. This observation suggests a primary role of the PKD protein family in the development of vascular integrity.

In the present study, we surveyed the exome of five case–parent trios with CCTs and identified ultrarare compound heterozygous variants in *PKD1L1* in two affected individuals. We investigated the impact of the identified variants on PKD1L1 protein localization in RPE1 cells. We further performed microscopy and microCT analysis of *Pkd1l1* mutant mouse embryos and performed immunofluorescence staining at 14.5 dpc of dorsal skin specimens of *Pkd1l1^+/+^*, *Pkd1l1^+/−^*, and *Pkd1l1^−/−^* littermates.

## 2. Materials and Methods

### 2.1. Exome Analysis in Five CCT Families

Exome sequencing (ES) was performed on 15 individuals in five families at the Cologne Center for Genomics (CCG), Cologne, Germany using methods previously described [25]. For the enrichment of genomic DNA, we used the NimbleGen SeqCap EZHumanExome Library v2.0 enrichment kit (Twist Bioscience HQ; 681 Gateway Blvd; South San Francisco, CA, USA). For ES, a 100 bp paired-end read protocol was used according to the manufacturer’s recommendations on an Illumina HiSeq2000 sequencer. Sequences were mapped to hg19 Genome Reference Consortium Human Build 37 (GRCh37), and QC was performed as previously described. Data analysis and filtering of mapped target sequences were performed with the “Varbank” exome and genome analysis pipeline v.2.1 as described previously. Further research on candidate genes prioritized by variant filtering was performed by phenotype analysis of existing knockout mice, expression in lymphoid tissue, functional domains hit by the respective variants, interaction with known CCT genes, and known gene–phenotype relations. The variants found in potential candidate genes were validated by Sanger sequencing, and segregation analysis was performed with all available family members. Detailed information on molecular details and clinical features for all five index families can be found in Appendix A. 

### 2.2. Pathogenicity Prediction Using SIFT, PolyPhen-2, and MutPred

The adverse effects of genetic variants of the human PKD1L1 were predicted using SIFT (https://sift.bii.a-star.edu.sg/) (accessed on 9 October 2023) [26,27]. SIFT uses the physical characteristics of amino acids and sequence homology to forecast whether alterations will be tolerated or harmful. For SIFT analysis, the chromosomal coordinates for every SNP were obtained from the dbSNP database. An online tool for annotating and coding nonsynonymous SNPs is PolyPhen-2 (Polymorphism Phenotyping-2) (http://genetics.bwh.harvard.edu/pph2/) (accessed on 9 October 2023) [28]. The program uses a particular empirical rule that considers physical and comparative factors in order to forecast the potential functional effects of an amino acid change on the composition and functionality of a human protein. PolyPhen-2 estimates the impact of a specific SNP or amino acid mutation at a given place in the query sequence using the query protein sequence in FASTA format as input. An online server called MutPred (http://mutpred.mutdb.org) was used to forecast the molecular causes of amino acid substitutions in mutant proteins that are linked to disease. It makes use of a number of characteristics linked to the evolution, structure, and function of proteins [29].

### 2.3. Three-Dimensional Structure Generation, Energy Minimization, and Superimposition

The three-dimensional structural models for wild-type PKD1L1 and mutants (G515R, V1282E, Q2183H, and N288Tfs*3) were created Using I-Tasser (http://zhanglab.ccmb.med.umich.edu/I-TASSER/ accessed on 8 November 2023). This program predicts protein 3D structures with nearly no manual intervention and uses an integrated combinatorial approach that combines the three standard conventional methods for structure modeling: comparative modeling, threading, and ab initio modeling [30]. After that, overall structural refinement and energy reduction were applied to all generated models using ModRefiner (http://zhanglab.ccmb.med.umich.edu/ModRefiner/) (accessed on 8 November 2023) [31,32]. This produced a model that was nearly identical to the original form in terms of side-chain placement, backbone topology, and H-bonding. The mutant proteins were superimposed onto an IL-8 wild-type protein, and the corresponding RSMD values were generated using SuperPose ver 1.0 (wishart.biology.ualberta.ca/Superpose/) (accessed on 8 November 2023) [33]. All functional motifs and domains in the PKD1L1 were predicted using Motif (https://www.genome.jp/tools/motif/) (accessed on 8 November 2023) [34].

### 2.4. Mouse Husbandry

Ethical approval for all mouse work was obtained from the UK Home Office, and experiments were carried out in accordance with the Medical Research Council (MRC) Harwell Ethics Committee. All mouse colonies were maintained in a pathogen-free environment at the Mary Lyon Centre, MRC Harwell Institute on a C3H/HeH background strain, where the *Pkd1l1^tm1Lex^* allele (*Pkd1l1^tm1^*) was created. Mice were housed in groups of 2–5 with controlled temperature (21 ± 2 °C) and humidity (55 ± 10%) in a 12 h light/dark cycle. Mice had free access to water and were fed ad libitum on a commercial diet (Special Diet Services, Essex, UK). Adult mice were sacrificed by cervical dislocation, whereas embryos were culled by decapitation or exsanguination in ice-cold PBS. The mouse embryos analyzed were a random mixture of males and females; their sex was not determined.

### 2.5. Mouse Genotyping

All mice for mating were genotyped at 3 weeks of age by collection of ∼1 mm diameter ear clips. DNA was extracted by adding 20 μg Proteinase K, 50 mM Tris pH 8.0, 0.5% Tween, 1 mM EDTA, and H2O to a total volume of 35 μL, followed by incubation at 55 °C for 1 h (for proteinase activity) and 95 °C for 5 min (to inactivate the enzyme). A total of 50 ng of extracted DNA was used in downstream genotyping assays. Allele counts were determined via quantitative reverse transcription PCR (RT-qPCR). Embryos were genotyped by collection of a tail segment or yolk sac after analysis and alleles were determined via PCR and amplicon visualization on agarose gels.

### 2.6. Mouse Embryo Phenotyping

After sacrificing the mother by cervical dislocation, embryos were removed and dissected in phosphate-buffered saline (PBS) under a light microscope. The total number of embryos in each litter was noted, as well as the presence or absence of embryos with gross edema. Each embryo was staged according to its phenotypic appearance, and the head was removed and the thoracic cavity opened. Embryos were scored according to lung lobation, heart apex position, heart outflow tract patterning, and stomach position. Any other gross abnormalities were also noted. All imaging was performed in PBS using a Teledyne Lumenera Infinity3-6URC camera on a Leica MZ12.5 microscope. Tail clips were taken after imaging for genotyping purposes.

### 2.7. Analysis of Lymphatic and Blood Vessels

Dorsal skin was collected from 14.5 dpc mouse embryos and fixed in 4% PFA in PBS overnight at 4 °C. Samples were washed twice in PBT (PBS + 0.2% Triton X-100) for 5 min at room temperature, then blocked for 2 h at room temperature in 10% heat-inactivated goat serum in PBT. Skin samples were incubated with primary antibodies in 10% heat-inactivated goat serum in PBT overnight at 4 °C, with gentle rotation. Capillaries were labeled using a primary antibody against endomucin (Abcam ab106100, 1/300, Cambridge, UK) and lymphatic vessels with Lyve1 (Abcam ab14917, 1/300). After, samples were washed thrice with 2% heat-inactivated goat serum in PBT for 15 min at room temperature, before incubating with secondary antibodies in 10% heat-inactivated goat serum in PBT for 2 h at room temperature, with gentle rotation. Secondary antibodies used were Goat antiRat DyLight 488 (Abcam ab96887, 1/1000) and Goat antiRabbit DyLight 594 (Abcam ab96885, 1/1000). Dorsal skin was then washed thrice with 2% heat-inactivated goat serum in PBT for 15 min at room temperature. Visualization of fluorescence was carried out using a Zeiss LSM710 microscope with Airyscan, and z-stacks were obtained. Maximum intensity projection of these z-stacks was used to generate composite images for analysis. Average vessel density, branching index, and diameter were calculated for each embryo using a minimum of 3 fields of view and 10 measurements per view. Analysis and preparation of images were performed using Zeiss Zen Blue version 2.3, AngioTool version 0.6a [35] and ImageJ release 1.52k. 

### 2.8. MicroCT

Embryos were dissected into ice-cold PBS, culled by exsanguination via the umbilical blood vessels, and the yolk sac was taken for genotyping. After washing with PBS, they were fixed in 4% PFA in PBS overnight (E14.5) or for 3 days (E15.5 and E16.5) at 4 °C, with gentle rotation. Embryos were stored in 1% PFA in PBS at 4 °C until ready for contrasting. Prior to contrasting, embryos were washed with PBS and transferred into a 50% Lugol solution (Sigma 32922). Incubation with this solution was performed at room temperature for 2 days (E14.5) or 1 week (E15.5 and E16.5), with gentle rotation. The Lugol solution was replaced every 2 days. 

### 2.9. Plasmid Construction

The full protein coding sequence for mouse Arl13b was amplified from testis cDNA and ligated into the pET-32a vector (Novagen 69015) at BamHI and HindIII sites. Ligation products were transformed into *Escherichia coli* DH5α and cultured at 37 °C with shaking, following a standard protocol. Transformants were selected using 100 µg/mL ampicillin in agar at 37 °C. Desired human *PKD1L1* sequence variants were incorporated into the pLenti-C-Myc-DDK-*PKD1L1* plasmid (Origene RC216376L1), which facilitates the production of lentiviral particles for expression of human PKD1L1 with C-terminal Myc and FLAG tags. To incorporate variants into this plasmid, recombination cloning was performed using the ClonExpress II One Step Cloning kit (Vazyme C112-02). Briefly, 1 µg of pLenti-C-Myc-DDK-*PKD1L1* was linearized using a unique restriction enzyme site close to the locus of the desired edit, before enzymes were heat-inactivated. Oligonucleotides containing the desired sequence variants were designed to contain homology arms of approximately 20 nucleotides in length that matched the plasmid at the site of linearization. These oligonucleotides were annealed, and recombination was performed following the manufacturer’s instructions. The products of recombination were transformed into *Escherichia coli* DH5α and subsequent culture at 30 °C with shaking, following a standard protocol. Transformants were selected using 25 µg/mL chloramphenicol in agar at 30 °C. All purified plasmid DNA was subjected to Sanger sequencing to confirm correct incorporation of the desired *PKD1L1* sequence variant or mouse Arl13b cDNA. 

### 2.10. Antibody Production

Recombinant N-terminal TrxA 6xHis-tagged mouse ARL13B protein was expressed in BL21(DE3)pLysS competent cells (Novagen) and purified by Ni-NTA affinity chromatography. Purified protein was injected into guinea pigs (by Covalab, Bron, France), and crude sera were used in immunostaining procedures.

### 2.11. Virion Production

HEK293T cells (ATCC CRL-3216) were cultured in DMEM (Gibco 31966-021, Waltham, MA, USA) with 10% FBS and plated in 10 cm dishes. Cells were allowed to become ~90% confluent prior to transfection. A total of 5 μg of pLenti-C-Myc-DDK-*PKD1L1* WT or variant was transfected per dish using the Lenti-vpak Lentiviral Packaging Kit (OriGene TR30037), following the manufacturer’s instructions. After 18 h, the culture medium was exchanged, and at 48 and 72 h post-transfection, the medium was collected and centrifuged at 600× *g* for 2 min. The supernatant was filtered through a 0.45 μm filter, and aliquots were stored at −80 °C until required. 

### 2.12. Cell Culture and PKD1L1 Protein Localization

hTERT RPE1 cells (ATCC CRL-4000) were cultured in DMEM/F-12 GlutaMAX (Gibco 10565-018) with 10% FBS in a 37 °C humidified cell culture incubator, 5% CO_2_ environment. Prior to transduction, the culture medium was exchanged and 10 μg/mL polybrene was added. Cells were seeded on coverslips in 6-well plates and transduced with lentiviral virions encoding human PKD1L1-Myc WT or variants when confluency reached approximately 70%. An MOI of 3 was used in all transductions. Then, 24 h post-transduction, the cell culture medium was exchanged for DMEM/F12 containing 0.5% FBS to stimulate ciliogenesis by serum starvation. Seventy-two hours post-transduction, cells were fixed in 4% PFA in PBS overnight at 4 °C, and immunofluorescence was performed using a standard protocol. 

Briefly, cells were washed twice with PBS and stained with DAPI for 5 min at room temperature. Permeabilization was performed using 0.2% Triton-X100 in PBS for 2 min at room temperature. After washing cells twice with PBS, samples were blocked for 30 min at room temperature in 3% BSA in PBS. Samples were then incubated with primary antibodies for 1 h at room temperature in 3% BSA in PBS. Primary antibodies against CEP164 (Santa Cruz Biotechnology sc-515403, 1/200, Santa Cruz, TX, USA), ARL13B (Harwell, 1/500), and Myc (Abcam ab9132, 1/300) were used. Cells were washed twice with PBS and incubated for 30 min at room temperature with fluorophore-conjugated secondary antibodies in 3% BSA in PBS. Secondary antibodies used were Goat antiMouse Alexa Fluor 647 (Abcam ab150119, 1/1000), Goat antiGuinea Pig Alexa Fluor 594 (Abcam ab150188, 1/1000) and Donkey antiGoat Alexa Fluor 488 (Abcam ab150133, 1/1000). Visualization of fluorescence was carried out using a Zeiss LSM710 microscope with Airyscan and z-stacks were obtained. Maximum intensity projection of these z-stacks was used to generate composite images for analysis. Analysis and preparation of images were performed using Zeiss Zen Blue software version 2.3. 

### 2.13. Individuals in the CCT Cohort

All five families included in this study provided informed consent according to the respective research protocols, approved by the Institutional Review Board of the University Hospital Bonn (No.152/18). Inclusion criteria for families were informed consent, primary CCT, and the absence of other conditions causing CCT, like congenital diaphragmatic hernia or major heart malformations. Blood or saliva samples were collected from affected individuals and unaffected parents. DNA was extracted from the blood and saliva samples using standard procedures.

## 3. Results

### 3.1. Putative Disease Variants Identified by Exome Survey

An exome survey (ES) of five families with CCT identified biallelic variants in *PKD1L1* in two independent families (Table 1). We did not identify any de novo variant or other biallelic recessive putative disease variants in any other gene (Appendix A). 

### 3.2. ES Results in Family 1

In Family 1 (CHT3), we identified compound heterozygous ultrarare *PKD1L1* missense variants (c.1543G>A(p.Gly515Arg) and c.3845T>A(p.Val1282Glu) in the affected segregating in the parents. Variant c.1543G>A(p.Gly515Arg) is novel according to gnomAD (https://gnomad.broadinstitute.org/), and variant c.3845T>A(p.Val1282Glu) was found to have an overall MAF of 0.0003 with no homozygotes reported. According to the ACMG criteria, we classified both variants as variants of uncertain significance (VUS). Variant c.1543G>A(p.Gly515Arg) was classified with the following criteria: PM2, PP3. Variant c.3845T>A(p.Val1282Glu) was classified with the following criteria: PP3. In the PKD1L1 protein structure variant, c.1543G>A(p.Gly515Arg) is located in the first functional polycystic kidney disease domain (PKD) of the protein, and variant c.3845T>A(p.Val1282Glu) is located in the functional receptor for egg jelly domain (REJ) of the protein (Figure 1).

A female infant was delivered by cesarean section as the first child of a first gravida with a gestational age of 33 + 3 weeks and a birth weight of 2900 g. Prenatally, she presented with left-sided pleural effusion with progressive dextro-positioning of the heart, hydrops fetalis, polyhydramnios, and a singular umbilical artery. She received three pleuroamniotic shunt insertions in utero. After birth, she was intubated and mechanically ventilated for 14 days; the pleural effusion was successfully treated with a postnatally placed left-sided pleural drain, which could be removed after eight days. Analysis of the pleural fluid led to the diagnosis of left-sided CCT. Sonographic examination of the brain, abdomen, and heart did not show any pathological findings. No infectious or any other cause of the hydrops fetalis could be found. She was discharged in good clinical condition after 32 days. 

### 3.3. ES Results in Family 2

In Family 2 (PUV146), we identified ultrarare compound heterozygous variants in *PKD1L1* in the affected segregating in the parents. The first variant constituted a novel loss-of-function variant (LoF) c.863delA(p.Asn288Thrfs*3) according to gnomAD. The second variant constituted a novel missense variant c.6549G>T(p.Gln2183His) according to gnomAD. According to the ACMG criteria, we classified the missense variant as VUS with the following criteria: PM2, PP3. Variant c.863delA(p.Asn288Thrfs*3) represents a LoF variant and was “Likely Pathogenic” with the following criteria: PVS1. In the PKD1L1 protein structure, variant c c.863delA(p.Asn288Thrfs*3) is located in the first extracellular domain of the protein, and variant c c.6549G>T(p.Gln2183His) is located in the fifth transmembrane domain of the protein (Figure 1).

A male infant was delivered by cesarean section as the second child of a fourth gravida with a gestational age of 33 + 2 weeks and a birth weight of 2500 g. Prenatally, he presented with bilateral pleural effusions and hydrops fetalis, which were treated with three pleuroamniotic shunt insertions in utero. After birth, he was intubated and mechanically ventilated; the pleural effusions were successfully treated with bilateral pleural drains. Due to his severe bilateral pulmonary hypoplasia, sufficient ventilation and oxygenation were not possible, and the patient died after one day from cardio-pulmonary failure.
Figure 1*PKD1L1* protein structure with variants in two unrelated CCT families: (**A**) PKD1L1 protein structure with functional domains, according to UniProt Q8TDX9; (**B**) compound heterozygous variants identified in two unrelated CCT families. The index patient of Family 1 presented with left-sided congenital chylothorax. The index patient of Family 2 presented with bilateral congenital hydrothorax. PKD, polycystic kidney disease domain; REJ, receptor for egg jelly domain; GPS, G protein-coupled receptor proteolytic site; t, transmembrane domain; PLAT, polycystin-1 lipoxygenase alpha-toxin domain; green, extracellular domains; yellow, cytoplasmatic domains.
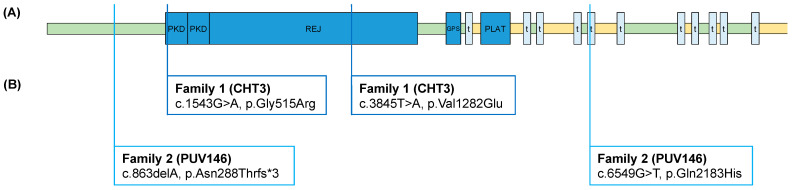


In silico predictions of the identified variants show pathogenic functional impact, especially for p.Gly515Arg and p.Asn288Thrfs*3 (Table 2). These two variants are also predicted to affect the functional domains of the PKD1L1 protein (Table 3). The other two variants were classified as possibly benign. 

Additionally, the residue lying between 514–516 also comprises a potential N-linked glycosylation site that has vanished due to the amino acid change at position 515. Predictions from sequence- and structure-based approaches were found to be in congruence with the structural modeling results. Structural modeling of the Pfam Motifs prediction showed that PKD1L1 possesses four distinct functional domains, REJ (aa713–933 and aa1139–1304), PKD (aa513–578, and aa600–662), polycystin domain (aa2442–2512), and the PLAT/LH2 domain (aa1798–1904) (Table 3). The PKD1L1 N288T mutant caused the frameshift resulting into gain of the stop codon and hence led to the formation of a short 288 amino acid length truncated protein devoid of all functional domains that are present in the wild type counterpart. Other two mutants, PKD1L1 V1282E and the PKD1L1 Q2183H were found to be neutral and non-pathogenic in the sequence and structure-based prediction approaches.

The three-dimensional structural models for wild-type PKD1L1 and mutants (Gly515Arg, Val1282Glu, Gln2183His, and Asn288Tfs*3) were created using I-Tasser (http://zhanglab.ccmb.med.umich.edu/I-TASSER/ accessed on 8 November 2023, respectively) (Table 4). This program predicts protein 3D structures with nearly no manual intervention and uses an integrated combinatorial approach that combines the three standard conventional methods for structure modeling: comparative modeling, threading, and ab initio modeling. After that, overall structural refinement and energy reduction were applied to all generated models using ModRefiner (http://zhanglab.ccmb.med.umich.edu/ModRefiner/ (accessed on 8 November 2023, respectively). This produced a model that was nearly identical to the original form in terms of side-chain placement, backbone topology, and H-bonding. The mutant proteins were superimposed onto an IL-8 wild-type protein, and the corresponding RMSD values were generated using SuperPose ver 1.0 (wishart.biology.ualberta.ca/Superpose/ accessed on 8 November 2023) (Table 5). All functional motifs and domains in the PKD1L1 were predicted using Motif (https://www.genome.jp/tools/motif/ accessed on 8 November 2023) (Figure 2).

### 3.4. Cellular Localization of Identified PKD1L1 Variants

Missense mutations in polycystin proteins can affect their ability to localize to cilia and result in phenotypic abnormalities [36]. To investigate whether the identified human PKD1L1 protein variants display normal subcellular localization to primary cilia, RPE1 cells were transduced with lentivirus encoding the normal wild-type full-length human *PKD1L1* protein-coding transcript or variants identified from the patients in this study. All except one variant (p.Asn288Thrfs*3) showed localization to the primary cilium no different from that observed for the WT protein, with puncta visible along the full length of the primary cilia (Figure 3A,B). For those variants, no other distinct subcellular localization was noted. The p.Asn288Thrfs*3 variant showed a ubiquitous presence within the cytosol of transduced RPE1 cells, and no obvious localization to cilia could be detected.

### 3.5. Incidence of Edema in Pkd1l1^−/−^ Mouse Embryos

Gross edemas have been reported in several mouse lines harboring homozygous mutations in members of the polycystin gene family, including *Pkd1*, *Pkd2*, and *Pkd1l1* [14,37,38]. The *Pkd1l1* allele previously associated with edema during embryonic development (Pkd1l1rks) is a missense mutation. We set out to establish the presence or absence of edema in *Pkd1l1^−/−^* embryos produced from the LoF Pkd1l1tm1 allele, which has not been previously noted. 

The phenotypes of *Pkd1l1^+/+^*, *Pkd1l1^+/−^*, and *Pkd1l1^−/−^* littermates were examined at 14.5, 15.5, and 16.5 dpc. These developmental time points were chosen due to previous observation of edema and situs defects at 14.5 dpc in the alternative Pkd1l1rks mouse line [14]. The frequency of each genotype from *Pkd1l1^+/−^* x *Pkd1l1^+/−^* matings was not considered to vary substantially from the expected frequencies; however, several dead *Pkd1l1^−/−^* embryos were identified at 14.5 and 15.5 dpc (Figure 4A). Approximately 30% of *Pkd1l1^−/−^* 14.5 dpc embryos displayed edema, with a reduced frequency at later time points (Figure 4B,C). Situs solitus, situs inversus, and situs ambiguous were found to be present in approximately equal proportions of *Pkd1l1^−/−^* embryos at 14.5 dpc, with varying changes to those frequencies at 15.5 and 16.5 dpc (Figure 4D,E). Between ~30–50% of *Pkd1l1^−/−^* embryos had some form of isomerism of the lungs, with left, right, and partial isomerisms observed (Figure 4F). Upon investigation of the heart, ~10–16% of *Pkd1l1^−/−^* embryos were found to have transposition of the great arteries (TGA) (Figure 4G). Stomach positioning was correct in 40–60% of *Pkd1l1^−/−^* embryos, with the remainder positioning the stomach on the right, consistent with randomized situs determination (Figure 4H). No situs defects or edema were noted in any *Pkd1l1^+/+^* or *Pkdl1l^+/−^* control littermates.

### 3.6. Loss of Pkd1l1 Can Result in Abnormal Lymphatic System Development

*Pkd1* and *Pkd2* LoF mouse embryos present aberrant lymphatic development, with greatly increased lymphatic vessel diameter, significantly reduced vessel density, and reduced branching [38]. These lymphatic defects are associated with the development of edema in the embryonic development of *Pkd1^−/−^* and *Pkd2^−/−^* mice. Since edema is also present in *Pkd1l1^−/−^* embryos, we hypothesized that similar to the aforementioned two members of the polycystin gene family, *Pkd1l1* may also have a role in normal lymphatic development. 

We collected dorsal skin from 14.5 dpc embryos and performed analysis via whole-mount immunofluorescence, using the lymphatic vessel marker Lyve1 to visualize lymphatic vessels and the venous marker endomucin to show capillaries. *Pkd1l1^−/−^* embryos displayed a range of lymphatic vessel abnormalities compared with their WT littermates, some of which were readily apparent when observing Lyve1 staining (Figure 4A,B). However, *Pkd1l1^+/−^* embryos displayed lymphatic and venous system development comparable to WT littermates. We quantified features such as vessel diameter, density, and branching, which revealed a higher degree of variation for lymphatic vessel density and branching across *Pkd1l1^−/−^* embryos compared with *Pkd1l1^+/+^* and *Pkd1l1^+/−^* littermates (Figure 5C,E). Contrastingly, this variation was not apparent for the density or branching of capillaries (Figure 5D,F). 

In *Pkd1l1^−/−^* embryos, the average lymphatic vessel diameter was highly varied across this genotype, (Figure 5G). While *Pkd1l1^+/+^* and *Pkd1l1^+/−^* embryos displayed mode average lymphatic vessel diameters of approximately 25–30 µm, *Pkd1l1^−/−^* embryos had fewer lymphatic vessels of this diameter, instead having an increased number of those with a diameter of >55 µm, which for the other genotypes was the upper limit observed (Figure 5H). Assessed separately, 42% (5/12) of *Pkd1l1^−/−^* embryos had a higher average or wider range of lymphatic vessel diameters compared with their WT littermates, 50% were similar to WT (6/12), and 8% (1/12) had an average lymphatic vessel diameter noticeably smaller than WT controls (Figure 5I).
Figure 4Phenotypes of Pkd1l1^−/−^ embryos: (**A**) Frequency of genotypes present at 14.5, 15.5, and 16.5 dpc resulting from Pkd1l1^+/−^ x Pkd1l1^+/−^ matings. (**B**) Percentage of embryos displaying external edema. (**C**) Edema present in exemplary Pkd1l1^−/−^ embryos. Pkd1l1^+/+^ littermates are shown for comparison. White arrows indicate the presence of edema in the thorax. (**D**) Percentage of each genotype displaying situs solitus (normal body patterning), situs inversus totalis (complete reversal), or situs ambiguous (abnormal but not complete reversal). (**E**) Photographs of situs observed at 14.5 dpc. Pkd1l1^+/+^ and Pkd1l1^+/−^ embryos display only situs solitus; however, Pkd1l1^−/−^ embryos present with either situs solitus, situs inversus totalis, or situs ambiguous. Asterisks indicate situs inversus totalis, white triangles indicate situs ambiguous, and images without shapes show situs solitus. Bottom Pkd1l1^−/−^ panel displays mild edema, left isomerism, and transposition of the great arteries (TGA). Black arrows indicate edema. The position of each lung lobe and the heart are labeled in the middle column in cases where they can be easily identified in situ. Right column shows lung lobation ex situ. H = heart, Cr = cranial lobe, Mi = middle lobe, Ca = caudal lobe, Ac = accessory lobe, Le = left lobe. (**F**) Lung lobation. Left, right, and partial isomerisms were observed in Pkd1l1^−/−^ embryos. (**G**) Heart positioning. Situs solitus and situs inversus, as well as transposition of the great arteries (TGA), were observed in Pkd1l1^−/−^ embryos. (**H**) Stomach positioning. Both situs solitus and situs inversus of the stomach were noted in Pdk1l1^−/−^ embryos.
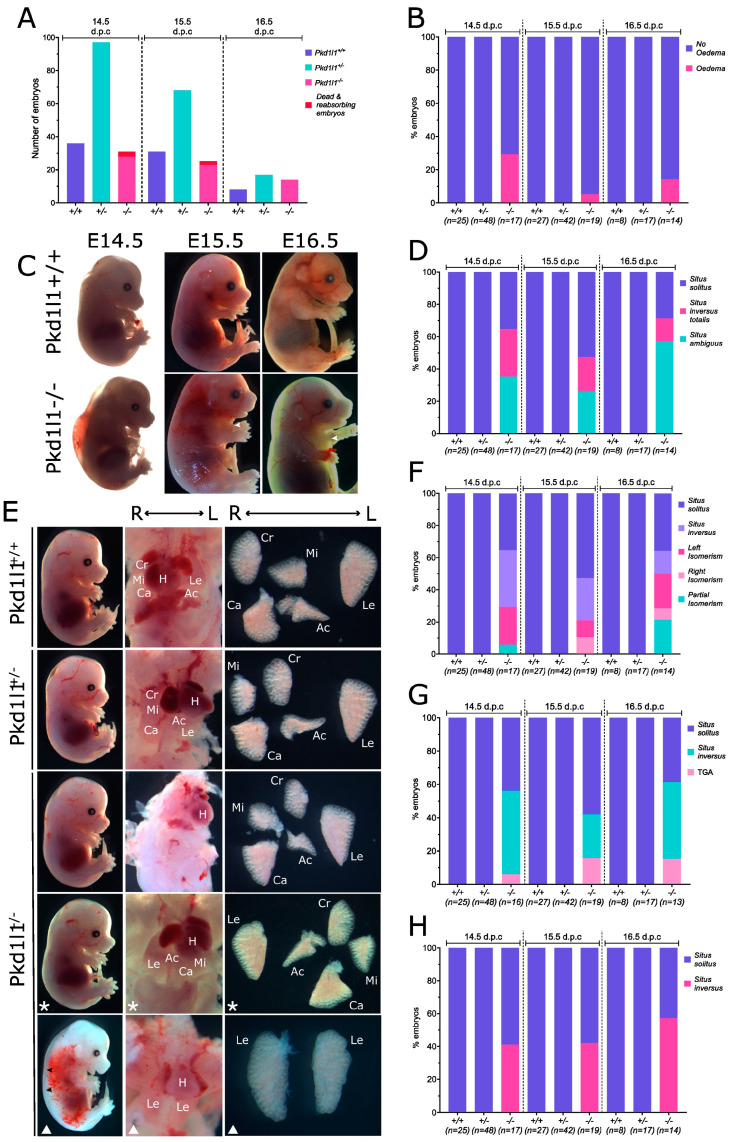

Figure 5Analysis of lymphatic vessels in embryonic dorsal skin: (**A**) Immunofluorescence of dorsal skin from 14.5 dpc embryos. Capillaries (red) were identified by positive immunostaining for endomucin and lymphatic vessels (green) are positive for Lyve1. Secondary antibody controls are shown for comparison. Pkd1l1^+/+^ and Pkd1l1^+/−^ show similar blood and lymphatic vessel morphologies; however, a range of normal and altered lymphatic vessel morphologies were observed in Pkd1l1^−/−^. (**B**) Representative examples of lymphatic vessel patterning in Pkd1l1^+/+^ vs. Pkdl1l^−/−^ dorsal skin samples, showing branchpoints (white asterisks) and several vessel diameter measurement points (orange lines). Scale bars = 200 µm. (**C**) Lymphatic vessel density in Pkd1l1 14.5 dpc embryo dorsal skin. Median, 25th, and 75th percentiles are shown. Average vessel densities were largely similar in all genotypes; however, a wider range was observed in Pkd1l1^−/−^ compared with littermate controls but did not reach statistical significance. (**D**) Capillary density in Pkd1l1 14.5 dpc embryo dorsal skin. Median, 25th, and 75th percentiles are shown. Pkd1l1^−/−^ samples do not appear to differ from their littermate controls. (**E**) Lymphatic vessel branching index in Pkd1l1 14.5 dpc embryo dorsal skin. Median, 25th, and 75th percentiles are shown. A wider range of branching indexes were observed in Pkd1l1^−/−^ compared with littermate controls but did not reach statistical significance. (**F**) Capillary branching index in Pkd1l1 14.5 dpc embryo dorsal skin. A wider range of branching indexes were observed in Pkd1l1^−/−^ compared with littermate controls but did not reach statistical significance. (**G**) Lymphatic vessel diameter in Pkd1l1 14.5 dpc embryo dorsal skin. A wider range of vessel diameters were observed in Pkd1l1^−/−^ samples, with some samples having vessels with diameters much wider than those observed in littermate controls but not reaching statistical significance. (**H**) Histogram of lymphatic vessel diameter measurements. Note that Pkd1l1^−/−^ overall had a lower number of vessel diameters centered around 30 μm and a higher number of vessel diameters > 50 μm. (**I**) Distribution of lymphatic vessel diameters in individual Pkd1l1 14.5 dpc embryos. The same measurements are plotted as in (**F**,**G**), with the median, 25th, and 75th percentiles shown. Five individual Pkd1l1^−/−^ embryos denoted by asterisks (*) displayed a wider range of vessel diameters, with a trend towards wider lymphatic vessels compared with littermate controls.
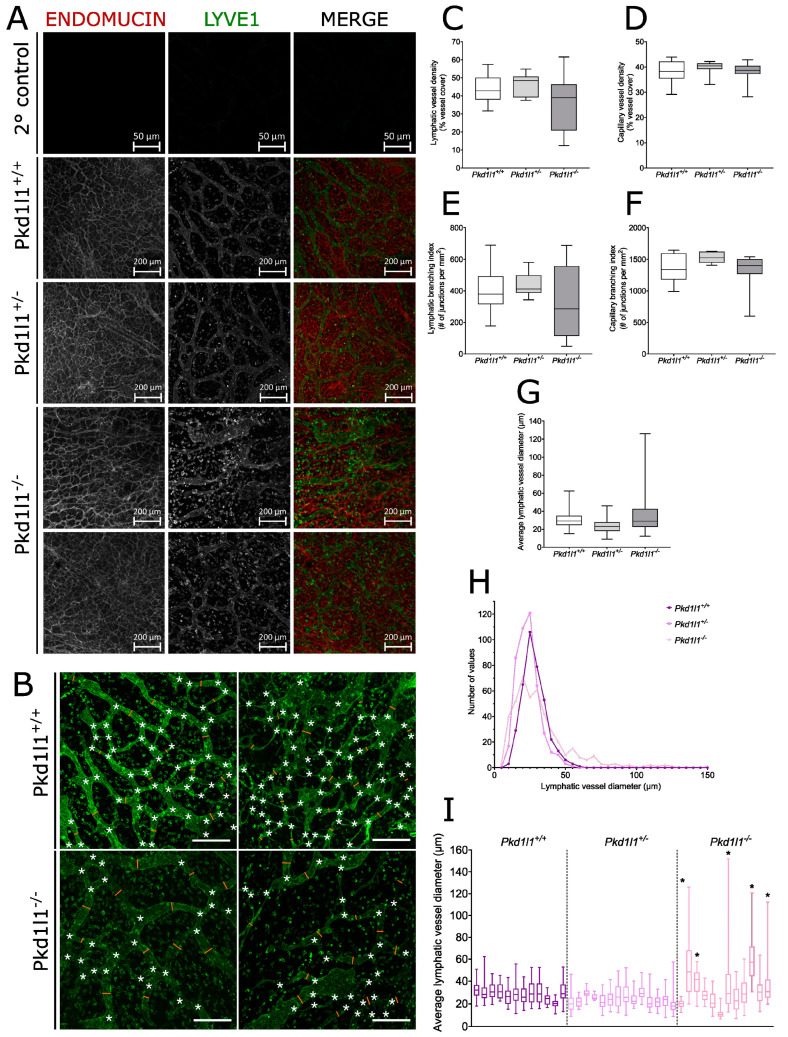


### 3.7. Observation of Pleural Effusion in Pkd1l1^−/−^ Embryos

The presence of pleural effusion is difficult to observe during the phenotyping of mouse embryos without the use of noninvasive technologies. We analyzed µCT data from 14.5 dpc embryos to determine if the edema present in *Pkd1l1^−/−^* embryos is primarily external or if they also present with pleural effusion. Strikingly, we observed the incidence of pleural effusion in several *Pkd1l1^−/−^* embryos (Figure 6). Fluid accumulation was noted in 2 of 14 embryos (14%). All control littermates were phenotypically normal. 

## 4. Discussion

To summarize, ES identified two unrelated families with biallelic compound heterozygous variants in *PKD1L1*. In Family 1 (CHT3), we detected compound heterozygous *PKD1L1* missense variants (p.Gly515Arg and p.Val1282Glu) in patient CHT3_501, who presented with left-sided CCT and hydrops fetalis. In Family 2 (PUV146), we detected compound heterozygous *PKD1L1* variants (p.Gln2183His and p.Asn288Thrfs*3) in patient PUV146_501, who presented with lethal bilateral hydrothorax and hydrops fetalis. 

In silico predictions of the identified variants show pathogenic functional impact, especially for p.Gly515Arg and p.Asn288Thrfs*3 (Table 2). The other variants are predicted to be less pathogenic. From this analysis, it appears that in order to have an embryonic and fetal impact on CCT formation in humans, it requires at least one variant that is predicted to be deleterious to some extent. These observations require further in vitro analysis of the variants’ impact on the development of the embryonic lymphatic vascular system.

Of the four *PKD1L1* variants identified, only two are located within characterized protein domains; p.Gly515Arg is located within the first PKD region and p.Val1282Glu is within the REJ domain. While the functional impact of all four variants is unknown at the molecular level, we have shown that with the exception of one (p.Asn288Thrfs*3), these protein variants can still localize to primary cilia, suggesting that subcellular mislocalization is not the main causative factor behind their pathogenicity. Distinct puncta were present along the length of cilia, characteristic of previous findings reported for murine PKD1L1 localization in the presence of PKD2 [14]. 

The diffuse subcellular pattern of p.Asn288Thrfs*3 localization implies a complete loss of or mislocalization of this PDK1L1 variant to the primary cilium. Several processes of protein localization to cilia have been identified, e.g., ciliary targeting sequences (CTSs) within proteins, requirements for post-translational modifications, and interactions with carrier proteins. Several CTSs have been reported for ciliary proteins, but there does not appear to be a conserved sequence; therefore, whether *PKD1L1* has an as-yet-unknown CTS remains to be seen. Nevertheless, the early truncation of *PKD1L1* in the p.Asn288Thrfs*3 variant does suggest the loss of key amino acid residues or protein regions crucial for the ciliary localization of PKD1L1. RPE1 cells are known to express PKD2 RNA, which raises the possibility of PKD2 protein translation and an interaction between PKD1L1 and PKD2 within RPE1 cells transduced with PKD1L1 [39]. Murine PKD1L! has been shown to localize to primary cilia when PKD2 is present but not alone, and their interaction is thought to be dependent upon the C-terminal coiled-coil domain in PKD1L1 [14]. The p.Asn288Thrfs*3 variant is a severe truncation that lacks the coiled-coil domain sufficient for ciliary localization. It remains unknown how the variants identified here influence the tertiary protein structure of PKD1L1 and their consequential influence on molecular interactions and downstream signaling pathways.

The majority of the situs defects we reported here in *Pkd1l1^−/−^* embryos, with the exception of edema and TGA, have been noted previously [36]. TGA is a developmental defect that has also been observed in embryos homozygous for a *Pkd1l1* point mutation (Pkd1l1rks) and in others homozygous for a *Pkd2* point mutation (Pkd2lrm4) [14,36]. While we cannot rule out the possibility of cardiac defects contributing to the onset of edema in the *Pkd1l1^−/−^* embryos used in this study, there is a higher proportion of embryos with edema compared with those with notable cardiac defects, suggesting alternative causation or combined factors. In some functional studies of genes associated with situs determination, including those of the polycystin family, there has been an assumption that edema in embryos is a result of structural cardiac defects [21]. Edema can be secondary to cardiac issues; however, it is unexpected to observe edema before birth in these cases [40]. Outeda et al. demonstrated in *Pkd1^−/−^* embryos that edema can occur in the absence of such defects [38]. Similarly, we also propose that the edema observed in our *Pkd1l1^−/−^* embryos is independent of any situs-associated cardiac structural abnormalities. 

*Pkd1^−/−^* and *Pkd2^−/−^* mouse embryos both display edema and lymphatic vessel abnormalities, similar to what we have shown here in *Pkd1l1^−/−^* mouse embryos, further corroborating the importance of the polycystin protein family in lymphatic development. We observed that overall, *Pdk1l1^−/−^* embryos presented a higher degree of variation in their lymphatic vessel densities and frequency of branching in dorsal skin than their *Pkd1l1^+/+^* or *Pkd1l1^+/−^* littermates; however, the median values were not significantly different. Most notably, however, a subset of *Pkd1l1^−/−^* embryos (42%) had lymphatic vessels that were considerably wider in diameter than controls and were easily identified as abnormal when viewed by fluorescence microscopy. We hypothesize that these malformed vessels have impaired functionality and therefore do not efficiently transport lymphatic fluid, contributing to lymph accumulation within tissues outside of lymphatic vessels, resulting in gross edema. In the present study, we were unable to define the dispersed LYVE1-positive cells in our dorsal skin samples as macrophages or pools of Lymphatic Endothelial Cells (LECs). The influence of PKD1L1 loss upon lymphatic precursor cells is still unknown; however, loss of PKD1 or PKD2 is known to cause their aberrant migration, and it is possible that this may also be true in the case of PKD1L1. Interestingly, the abnormal vessel features we observed were limited to those of a lymphatic nature, with capillaries being unaffected. This would suggest that PKD1L1 is important for normal lymphatic system development but has a limited or no role in the development of the peripheral vasculature system outside of the situs.

The presence of pleural effusion in *Pkd1l1^−/−^* embryos is of great interest due to the identification of human *PKD1L1* variants associated with pleural effusion in this study. A thorough analysis of heart structure was not performed; therefore, we cannot rule out the possible presence of unidentified cardiac or circulatory system defects that may be contributing to the pleural effusion in our *Pkd1l1^−/−^* embryos. Composition of the accumulated fluid from these embryos was not performed; therefore, a definitive conclusion about chylothorax is beyond the scope of this study. However, in conjunction with our findings on the abnormal lymphatic vessels and external edema, the observation of pleural effusion indicates that Pkd1l1 is an important mediator of lymphatic system development in both mice and humans, suggesting a conserved function across these two species. Compound heterozygous or homozygous LoF *PKD1L1* variants can result in pleural effusion with or without changes to situs. This study expands our understanding of *PKD1L1* during embryonic development and highlights this gene as one of importance when investigating CCTs.

## 5. Conclusions

We suggest *PKD1L1* as a novel recessive candidate gene for CCTs. There is evidence in both humans and mice that PKD1L1 contributes to lymphatic system development.

## Figures and Tables

**Figure 2 cells-13-00149-f002:**
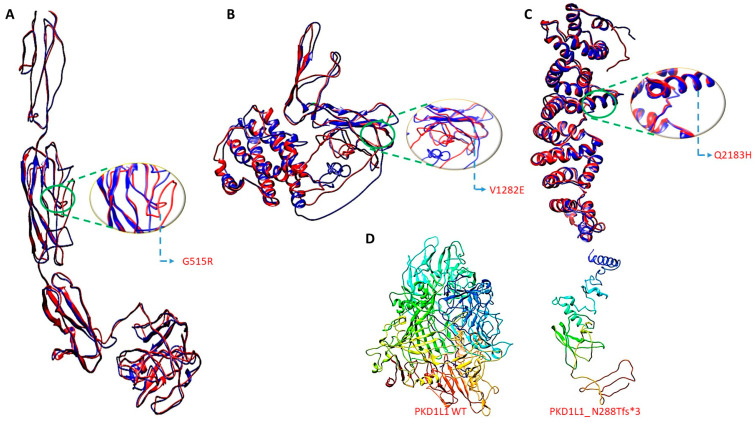
Three-dimensional structure generation, energy minimization, and superimposition. Three-dimensional structure generation, energy minimization, and superimposition. (**A**) PKD1L1 mutant Gly515Arg superimposed onto PKD1L1 wildtype, (**B**) PKD1L1 mutant Val1282Glu superimposed onto PKD1L1 wildtype, (**C**) PKD1L1 mutant Gln2183His superimposed onto PKD1L1 wildtype, and (**D**) PKD1L1 wildtype and its truncated PKD1L1 Asn288Tfs*3 lacking functional domains.

**Figure 3 cells-13-00149-f003:**
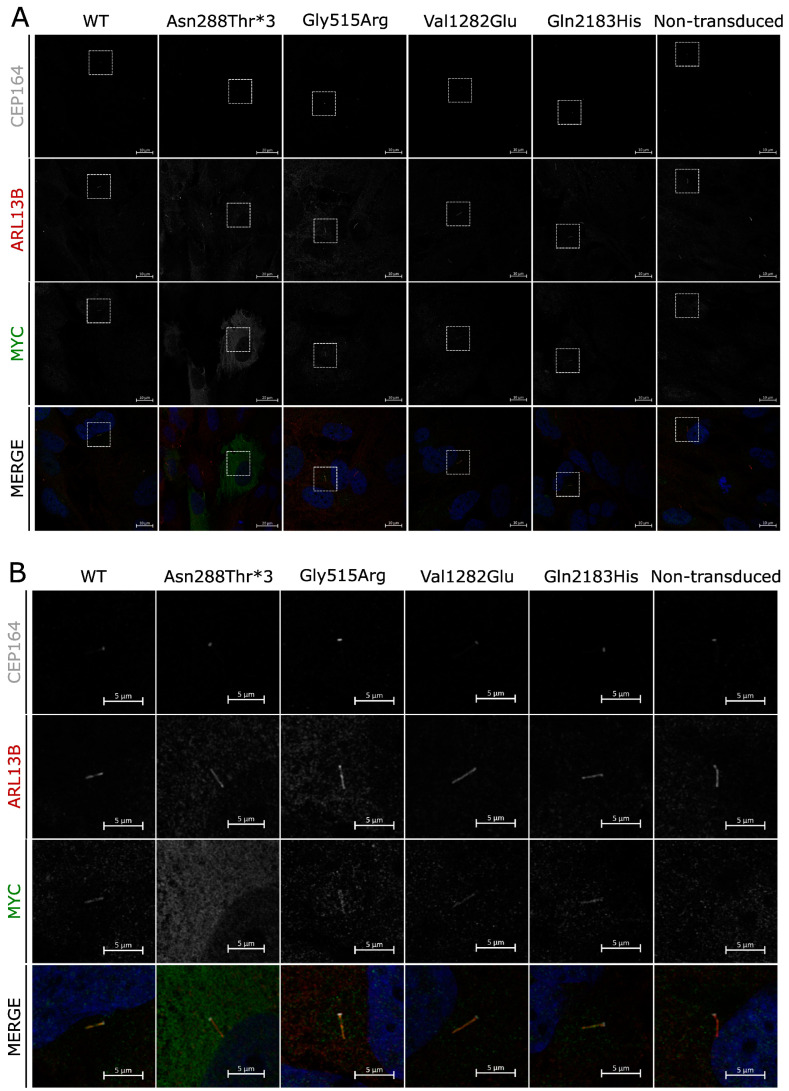
Subcellular localization of *PKD1L1* variants: (**A**) Immunofluorescence of RPE1 cells transduced with virions facilitating expression of human PKD1L1-MYC protein variants. CEP164 (gray), ARL13B (red), and MYC (green) are shown separately and merged with DAPI (blue) for each variant, alongside WT and nontransduced controls. For each variant, subset regions of interest are depicted (white boxes). (**B**) Regions of interest chosen from section A of this figure, each depicting the primary cilium of a transduced cell. All variants localize to primary cilia, with the exception of Asn288thr*3, which appears diffuse throughout the cytoplasm.

**Figure 6 cells-13-00149-f006:**
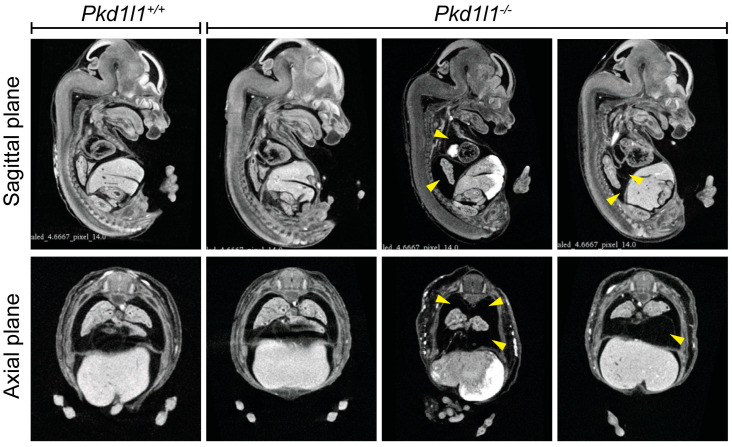
Pleural effusion in Pkd1l1^−/−^ embryos: Sagittal and axial plane views of 14.5 dpc embryos assessed by µCT. Pleural effusion was noted in two Pkd1l1^−/−^ embryos (2/14, 14%) but not in control littermates (14 Pkd1l1^+/+^, 15 Pkd1l1^+/−^ examined). Yellow arrows indicate regions of fluid accumulation.

**Table 1 cells-13-00149-t001:** Molecular Details and Clinical Features of Individuals with Variants in *PKD1L1*.

		Family 1 (CHT3)	Family 2 (PUV146)
Molecular Details	Individual	II-1	II-1
	Zygosity	Compound Heterozygous	Compound Heterozygous
	Inheritance	Autosomal Recessive	Autosomal Recessive
	Variant location *PKD1L1* (NM_138295.4)	c.1543G>A, p.Gly515Arg c.3845T>A, p.Val1282Glu	c.863delA, p.Asn288Thrfs*3 c.6549G>T, p.Gln2183His
	Variant consequence	Missense Missense	Frameshift Missense
	gDNA location Chr7 (NC_000007.13)	g.47944902C>T g.47913548A>T	g.47968998delT g.47869647C>A
	Exon	11/57 24/57	7/57 43/57
	gnomAD MAF	Not Reported 0.0003 (hom = 0)	Not Reported Not Reported
	Polyphen-2	Probably Damaging Benign	N/A Benign
	SIFT	Tolerated Tolerated	N/A Tolerated
	Mutation Taster	Disease Causing Polymorphism	N/A Polymorphism
	CADD	29.0 7.758	N/A 23.6
	ACMG Criteria	VUS (PM2, PP3) VUS (PM2)	Likely pathogenic (PVS1, PM2) VUS (PM2)
Clinical Features	Sex	Female	Male
	Age of Onset	Congenital	Congenital
	Primary Phenotype	Chylothorax, left Hydrops fetalis	Hydrothorax, bilateral Hydrops fetalis
	Secondary Phenotype	Persistent pulmonary hypertension, respiratory failure	Severe pulmonary hypoplasia, persistent pulmonary hypertension, cardio-respiratory failure
	Prenatal intervention	3x shunt insertion, thoracocentesis	3x shunt insertion

* Hom, reported number of homozygotes; N/A, not applicable; VUS, variant of uncertain significance.

**Table 2 cells-13-00149-t002:** In silico pathogenicity predictions for the *PKD1L1* variants identified by ES.

Mutant	SIFT		POLYPHEN-2		MutPred		Overall Functional Impact
	Prediction	Score	Prediction	Score	Prediction	Score	
p.Gly515Arg	Deleterious	0.02	Probably damaging	0.95	Severe Functional Impact	0.626	Pathogenic
p.Val1282Glu	Neutral	0	Benign	0.017	Less Impact	0.482	Less pathogenic
p.Gln2183His	Neutral	0	Benign	0.15	Less Impact	0.343	Less pathogenic
p.Asn288Thrfs*3	-	-	-	-	-	-	Pathogenic

**Table 3 cells-13-00149-t003:** Functional domains and in silico pathogenicity predictions for the *PKD1L1* variants identified by ES.

Pfam Motifs	AA Position	Description	Independent E-Value	WT	Gly515Arg	Val1282Glu	Gln2183His	Asn288Thrfs*3
REJ	713–933	PF02010, REJ domain	3.70 × 10^−18^	benign	benign	benign	benign	pathogenic
	1139–1304		1.20 × 10^−16^	benign	pathogenic	benign	benign	pathogenic
PKD	513–578	PF00801, PKD domain	6.90 × 10^−6^	benign	pathogenic	benign	benign	pathogenic
	600–662		1.60 × 10^−11^	benign	benign	benign	benign	pathogenic
Polycystin_ dom	2442–2512	PF20519, Polycystin domain	1.40 × 10^−15^	benign	benign	benign	benign	pathogenic
PLAT	1798–1904	PF01477, PLAT/LH2 domain	5.20 × 10^−14^	benign	benign	benign	benign	pathogenic
PKD_4	618–661	PF18911, PKD domain	4.00 × 10^−6^	benign	benign	benign	benign	pathogenic
PKD_channel	2518–2662	PF08016, Polycystin cation channel	1.20 × 10^−5^	benign	benign	benign	benign	pathogenic
Glyco_trans _2_3	2584–2676	PF13632, Glycosyl transferase family group 2	0.013	benign	benign	benign	benign	pathogenic
Ion_trans	2528–2738	PF00520, Ion transport protein	0.11	benign	benign	benign	benign	pathogenic

**Table 4 cells-13-00149-t004:** The statistical results of the 3D structure model of various mutant PKD1L1 proteins using I-Tasser.

Structural Modeling	Alignment			Best Model Score		
	PDB Matched	Coverage (%)	Normalized Z-Score	C-Score	Estimated TM-Score	Estimated RMSD (Å)
PKD1L1_Gly0515_WT	4m00A	96	1.05	−0.97	0.59 ± 0.14	09.1 ± 4.6
PKD1L1_Arg0515_MT	4m00A	96	1.05	−0.94	0.60 ± 0.14	09.1 ± 4.6
PKD1L1_Val1282_WT	7z01A	88	1.27	−2.60	0.41 ± 0.14	12.8 ± 4.2
PKD1L1_Glu1282_MT	7z01A	86	1.28	−3.00	0.37 ± 0.13	13.9 ± 3.9
PKD1L1_Gln2183_WT	6a70	73	6.40	−0.71	0.62 ± 0.14	08.2 ± 4.5
PKD1L1_His2183_MT	6a70	74	4.66	−0.89	0.60 ± 0.14	08.6 ± 4.5
PKD1L1_Asn288Tfs*3				−0.97	0.59 ± 0.14	9.1 ± 4.6

**Table 5 cells-13-00149-t005:** The changes in the local RSMD and the global RSMD for the various mutant PKD1L1 predicted using SuperPose ver 1.0.

Superimposing		Local RMSD (Å)		Global RMSD (Å)	
		Alpha-C	Backbone	Alpha-C	Backbone
PKD1L1_Gly515_WT	PKD1L1_Arg515_MT	0.82	1.11	5.97	5.96
PKD1L1_Val1282_WT	PKD1L1_Glu1282_MT	0.71	0.86	9.18	9.23
PKD1L1_Gln2183_WT	PKD1L1_His2183_MT	2.82	2.84	2.82	2.84

## Data Availability

The data that support the findings of this study are available on request from the corresponding author. The data are not publicly available due to privacy or ethical restrictions.

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
