# Peer review of "PKD1L1* Is Involved in Congenital Chylothorax"

_cells, 2024, doi:10.3390/cells13020149_

Round 1
Reviewer 1 Report
Comments and Suggestions for Authors
I did not find a lot of problems with the language. The language could be improved in some places to make it sound less redundant.
Author Response
Reviewer #1:
Whitchurch & Schneider et al. in the study have studied the potential role of PKDIL1 in lymphatic development and its mutation in lymphatic anomalies, in this case Congenital Chylothorax (CCT). The function of lymphatic vessels has been well studied in the past decades. One of the conditions affecting lymphatic vessel structure and function are lymphatic anomalies. Scientists have struggled to identify the mutations that contribute to several lymphatic anomalies including lymphedemas. Identification of genetic mutations in congenital lymphatic anomalies have led scientists to discover function of these genes in the development of lymphatic vasculature. Hence, identification of mutations in new genes on context to lymphatic anomaly helps the field of lymphatic development to move forward. Authors in the study have identified variants in PKD1L1 gene and suggested its role in the development of lymphatic system because of the lymphatic anomaly in the embryos deficient of Pkd1l1. While the study provides several interesting information, and certainly is of high interest to the field of lymphatic biology. However, there are several missing links and claims in studies needs to be shown by further evidence. Please find below specific comments.
Major comments
Comment 1:
Resolution of Figure 2 is too low to see any information in the image. Authors should change the resolution to make it presentable in the manuscript. High magnification image of the cilia should be taken to show any differences like mentioned in discussion for the distinct puncta (Line 468).
Answer to Comment 1:
The inset images of cilia have been enlarged and file type during figure assembly has been changed to preserve better quality images. Unfortunately, due to technical and sample limitations we cannot repeat these immunofluorescence experiments to provide better resolution.
Comment 2:
In Figure 3E: It would be better if authors label organs position in the middle column of the figure to have better sense of anatomical positioning of the organ in the entire embryos.
Answer to Comment 2:
Thank you for this comment – this has now been implemented into the figure.
Comment 3:
Figure 4: There are lot of problems with this part of the manuscript. This Figure should be worked and needs extensive attention. Figure 4B and C, Figure 4D and E are exactly mirroring image of each other. However, in the Figure legend authors have labelled as C and D for capillary density. I am not sure if I understand this. Again, the contrast of the image is poor and difficult to observe the vessels clearly. Also, author should use combination of lymphatic markers like PROX1 and NRP2 which is a standard use in the field. There are lot of scattered green (LYVE1) stained cells (based on what it looks). Are they just LYVE1+ macrophages or Lymphatic Endothelial Cells (LECs) pools dispersed around? It is always advised to use combination of markers including PROX1, NRP2 and LYVE1 at this embryonic stage. Using combination of markers should help understand the architecture and information better. Authors should also have high resolution image to show the branch points clearly and the diameters since they look clearly different in the Pkd1l1-/-mutants. The image should be taken in a way along the midline of the embryonic skin. This kind of imaging should provide information of lymphatic coverage, branchpoint, diameter, sprouting etc. This would also help demarcate the regions that are. For reference: Chiang et al., 2023; The EMBO Journal.
Answer to Comment 3:
Thank you for bringing the issues regarding Fig. 4B, C, D & E to our attention. These have now been corrected to show the correct data as described in the figure legend. We have also added and labeled higher resolution images to indicate several measured diameters and branchpoints, making it clearer to the reader what has been analysed and demonstrating the clear differences between genotypes. We appreciate the guidance regarding the use of lymphatic markers recommended in the field, however due to technical and sample limitations we are unable to repeat these immunofluorescence experiments as suggested by the reviewer. We will take these helpful comments on board for future investigations. Text in the manuscript has been modified to comment on the scattered LYVE1 positive cells and clarify that we unable to determine if they are microphages or LECs in this study.
Minor comments:
Comment 1:
Line 274, 288, 296: Wrong Figure citation. Should be Figure 1 instead of Figure 4.
Answer to Comment 1:
We have corrected the wrong Figure citation accordingly. Since we added an additional Figure, a new Figure 2, showing the “three-dimensional structure generation, energy minimization and superimposition of WT PKD1L1 protein and mutated PKD1L1 proteins”, Figure citations were changed accordingly, including a new Figure 3A and 3B.
Comment 2:
Please remake the table to make it easy to follow. Molecular Details and Clinical features segment could be differently coloured.
Answer to Comment 2:
We revised the table and added colouring of the segments accordingly.
Comment 3:
Figure 3A: It would be nice if the author can mention the significance of colour already here rather than in Figure B. Label for only Dead and reabsorbing embryos is shown in Figure 3A.
Answer to Comment 3:
This figure has been updated to show the correct colours in the key, which were previously absent.
Comment 4:
Figure 4G, Label in Y axis says number of valves. I do not see any valves being quantified here. Please be careful with consistency in labels and Figure legends to makes sure they align with the manuscript findings.
Answer to Comment 4:
This label says “values” rather than “valves”, but we have changed the text size and layout to make to make axis labels easier to read.
Comment 5:
Does the authors have any information on the patient’s lymphatic function. Did the affected patients have any edema in lower or upper extremities. Also, were the lymphoscintigraphy done in these patients?
Answer to Comment 5:
The authors HR and AM where the primary care takers during the neonatal period of the described patients. Neither of the described patients had any cutaneous edema in their neonatal period comparable to typical lymphedema seen in patients with mutations in VEGFR3 associated Milroy disease. Clinical consequences associated with classic nonsyndromic CCT manifesting prenatally are usually mass effects that result in pulmonary hypoplasia, decreased pre-load, heart failure and non-immune hydrops fetalis (doi:10.3389/fped.2021.633051). Hence, in these cases clinicians do not seek diagnostic lymphoscintigraphy. Lymphoscintigraphy in our unique German Centre for Congenital Lymphatic Vascular Diseases (https://zseb.ukbonn.de/b-zentren/seltene-paediatrische-erkrankungen/angeborene-lymphgefaesserkrankungen/) is carried out on a regular bases for chylothoraces occurring in association with clinical syndromes not responding to dietary treatment in the neonatal period or when CCTs occur later during the first year of life with the suspicion of lymphatic vascular disease. However; lymphoscintigraphy is not recommended or carried out for nonsyndromic CCT responding to standard neonatal treatment.
Comment 6:
Line 396: There are no capillary quantifications.
Answer to Comment 6:
Please see answer to comment 3 – capillary quantification graphs have now been added at the correct positions.
Comment 7:
Did the authors check the level of PKD1L1 in the lymphatic endothelial cells( LECs) versus blood endothelial cells? Or are there any published evidence PKD1L1 expression in LECs.
Answer to Comment 7:
In the present study, we did not investigate the expression level of PKD1L1 in lymphatic endothelial cells (LECs), which is the focus of our follow-up project currently funded by the German Research Foundation (Deutsche Forschungsgemeinschaft (DFG) - Project number 517059954). However, previous publicly available expression data has found PKD1L1 to be expressed lymph nodes but has not been mentioned for LECs (https://www.bgee.org/gene/ENSG00000158683; https://www.proteinatlas.org/ ENSG00000158683-PKD1L1/tissue/lymph+node). We therefore assume that the involvement of PKD1L1 in CCT might not be due to direct impact on the function of LECs but rather on the interaction of PKD1L1 with PKD1 and/or PKD2. As PKD1L1 belongs to the polycystin family it has significant homology with all known polycystins, but the longest stretches of homology were found with polycystin-1, PKD1. Interestingly, PKD1L1 represents a PKD2-interacting protein (DOI: 10.1371/journal.pgen.1006070). In this context, Outeda et al. (https://doi.org/10.1016/j.celrep.2014.03.064) found both, PKD1 and PKD2 to be involved and expressed in mouse precursor LECs. They found that mice with germline deletion of Pkd1 or Pkd2 develop severe and lethal edema in midgestation likely to be caused by defects in lymphatic vascular development. Pkd1 or Pkd2 mutant embryos exhibited reduced lymphatic vessel density and vascular branching along with aberrant migration of early LEC precursors. While their work does not directly imply PKD1L1 in maldevelopment of LECs nor an expression of PKD1L1 in LECs, their work may suggest a possible link between mutated PKD1L1 protein and secondary dysfunctional interaction with PKD1 and/or PKD2.
Comment 8:
Did the authors check the early embryonic time points of Pkd1l1 mice, especially around Lymphatic specification (E9.5) and later when cells bud off from the cardinal vein? Did they observe any effect in early time points?
Answer to Comment 8:
Regretfully, we did not check earlier time points as part of this study.
Comment 9:
While Figure 4 shows that lymphatic vessels are affected in Pkd1l1 embryos, it does not fully provide evidence of how lymphatic vessels are affected. Authors should study the structure of lymphatic vessels in Pkd1l1 embryonic skin in more details since same materials could be used to collect enough information that should provide information on whether LECs proliferation or migration or any other affects are observed. Ideal approach is to use dorsal midline and use low and high resolution images to extract quality information for quantifications.
Answer to Comment 9:
We greatly appreciate the reviewer’s suggestion of how to acquire more detailed information for lymphatic vessel development in these samples. Unfortunately, due to technical and sample limitations, we are unable to reexamine these dorsal skin samples and collect additional data.
Comment 10:
Authors have identified compound heterozygous variants in PKD1L1 patients. Figure 2 answers if individual variation has effect on the protein localization and expressions. However, since these are compound heterozygous variations, author should use those variant combinations seen patients together in cells to study in one of the variants acts as a modifier for the other variants. Would you see the additive effect in the presence of both variants?
Answer to Comment 10:
We thank the reviewer for this comment. Certainly, in order to investigate if the variants in biallelic composition have an effect on ciliary function, we would have needed to test them in combination in a functional assay. The presented investigation however intended to investigate whether the identified human PKD1L1 protein variants display normal subcellular localization to primary cilia, since we were previously able to show, that missense variants in Polycystin proteins can affect their ability to localize to cilia (doi: 10.1371/journal.pgen.1006070). Here, we found all except one variant (p.Asn288Thrfs*3) to localize to the primary cilium no different from that observed for the WT protein, with puncta visible along the full length of primary cilia. For those variants, no other distinct subcellular localization was noted. The p.Asn288Thrfs*3 variant showed a ubiquitous presence within the cytosol of transduced RPE1 cells and no obvious localization to cilia could be detected.
Comment 11:
Author have used RPE1 cells for the variant expression study, which is biased by the possibility of the presence of PKD2 in these cells to modify the effect of the mutants. To rule out whether the effect is because of PKD2 presence, the authors should study the cells that are void of PKD2.
Answer to Comment 11:
This is a fair comment; however it is beyond the scope of this study to assess PKD1L1 protein variant subcellular localisation +/- PKD2. This could be addressed in future work for a more detailed investigation.
Final remark of Reviewer #1
To conclude, I find the study very interesting and am confident that the findings should contribute significantly to the existing understanding in the field. However, there are a lot of loose ends, missing information, badly done imaging and figure extractions which should be improved in order to make this interesting finding to the publication.
Answer to final remark of Reviewer #1:
We thank the reviewer for his comments and thoughts. They have improved the quality of our manuscript and we hope to have answered all queries satisfactory.

Reviewer 2 Report
Comments and Suggestions for Authors
The present study reports new data on the involvement of PKD1L1 in congenital chylothorax. This is a very nice comprehensive study that adds clinical data on the exome analysis of ultra-rare compound heterozygous variants in case-patients. The novelty of this study relies on the discovery of 2 case-parents with CCT and the analysis of the PKD1L1 variants associated with CCT. Overall the study is very interesting for the field. The protein localization of PKD1L1 and its relationship with lymphatic vessel abnormalities were reported using cell models and Pkd1L1-/- embryos. Methodology and data analysis are well sounded however the quality of the images could be improved. Although it is possible to correlate a protein dysfunction with the lymphatic alteration, more studies could be included to discard confounding results related with heart diseases, inflammatory reaction or hypoxia/cellular death both in cell and animal models. The study lacks information regarding the involvement of downstream pathways (RAS/MAPK or PI3K/AKT) in the PKD1L1 mislocalization. It would also be interesting to understand how these variants have been transmitted in these families as well as potential alleles carriers with less obvious symptoms.
Comments on the Quality of English LanguageOverall the manuscript is well written, however it could be improved looking for typos and some grammatical errors.
Author Response
Reviewer #2:
Comment 1:
The present study reports new data on the involvement of PKD1L1 in congenital chylothorax. This is a very nice comprehensive study that adds clinical data on the exome analysis of ultra-rare compound heterozygous variants in case-patients. The novelty of this study relies on the discovery of 2 case-parents with CCT and the analysis of the PKD1L1 variants associated with CCT. Overall the study is very interesting for the field. The protein localization of PKD1L1 and its relationship with lymphatic vessel abnormalities were reported using cell models and Pkd1L1-/- embryos. Methodology and data analysis are well sounded however the quality of the images could be improved. Although it is possible to correlate a protein dysfunction with the lymphatic alteration, more studies could be included to discard confounding results related with heart diseases, inflammatory reaction or hypoxia/cellular death both in cell and animal models. The study lacks information regarding the involvement of downstream pathways (RAS/MAPK or PI3K/AKT) in the PKD1L1 mislocalization. It would also be interesting to understand how these variants have been transmitted in these families as well as potential alleles carriers with less obvious symptoms.
Answer to Comment 1:
We want to thank Reviewer #2 for his comments. From our analysis we can conclude, that co-occurring heart diseases did not correlate with the occurrence of CCT respectively general edema in the Pkd1L1-/- mice embryos. We assume that the occurrence of CCT and general edema these mice embryos is not related to hemodynamical impairments.
To the best of our knowledge, PKD1L1 protein is not involved in downstream signaling of RAS/MAPK or PI3K/AKT pathways (see String Protein Version 12.0 analysis on PKD1L1, December 18th 2023). PKD1L1 protein (human) - STRING interaction network (string-db.org)
Previously, we were able to show, that PKD1L1 is implicated in embryonic left-right (L-R) patterning. In this we were able to show, that PKD1L1 protein can mediate a response to flow coheres with a mechanosensation model of flow sensation in which the force of fluid flow drives asymmetric gene expression in the embryo https://doi.org/10.1371/journal.pgen.1006070).
Comment 2:
Overall the manuscript is well written, however it could be improved looking for typos and some grammatical errors.
Comments on the Quality of English Language
We have revised the manuscript accordingly and erased typos and grammatical errors.
